# NDCN-Brain: An Extensible Dynamic Functional Brain Network Model

**DOI:** 10.3390/diagnostics12051298

**Published:** 2022-05-23

**Authors:** Zhongyang Wang, Junchang Xin, Qi Chen, Zhiqiong Wang, Xinlei Wang

**Affiliations:** 1School of Computer Science & Engineering, Northeastern University, Shenyang 110819, China; wangzhongyang@cse.neu.edu.cn (Z.W.); wangxinlei@stumail.neu.edu.cn (X.W.); 2Key Laboratory of Big Data Management and Analytics, Northeastern University, Shenyang 110819, China; 3College of Medicine and Biological Information Engineering, Northeastern University, Shenyang 110819, China; 2071212@stu.neu.edu.cn (Q.C.); wangzq@bmie.neu.edu.cn (Z.W.)

**Keywords:** dynamic network, extension, NDCN, fMRI, cognitive impairment diseases

## Abstract

As an extension of the static network, the dynamic functional brain network can show continuous changes in the brain’s connections. Then, limited by the length of the fMRI signal, it is difficult to show every instantaneous moment in the construction of a dynamic network and there is a lack of effective prediction of the dynamic changes of the network after the signal ends. In this paper, an extensible dynamic brain function network model is proposed. The model utilizes the ability of extracting and predicting the instantaneous state of the dynamic network of neural dynamics on complex networks (NDCN) and constructs a dynamic network model structure that can provide more than the original signal range. Experimental results show that every snapshot in the network obtained by the proposed method has a usable network structure and that it also has a good classification result in the diagnosis of cognitive impairment diseases.

## 1. Introduction

At present, the clinical diagnosis of neurodegenerative diseases is mainly based on the scale method, which can quickly evaluate and screen the severity of the patient’s disease, but it is not sensitive enough for the early stage of disease [1,2]. For these reasons, neuroimaging methods are mostly supplemented in the study of changes in brain functional patterns, such as positron emission tomography (PET), structural magnetic resonance imaging (sMRI), and functional magnetic resonance imaging (fMRI). Among them, PET requires contrast agents, which are harmful to the body; sMRI is suitable to capture microscopic levels of neuronal loss and gray matter atrophy; however, not all neurodegenerative diseases produce these changes. The neuroimaging data represented by fMRI based on blood oxygenation level dependent (BOLD) have the advantages of being non-invasive, repeatable, and high spatiotemporal resolution, which not only has an excellent display effect on brain structure, but can also reflect the timing changes of the brain’s functional status [3]; therefore, it is an ideal method to study the activity patterns and the relationships across the brain. Moreover, it provides new insight into the study of the functional status of the brain in patients with neurodegenerative diseases. Modern brain imaging techniques and statistical physics, especially complex network theory, provides the necessary basis and analysis methods for the study of human brain functional networks (BFN) using neuroimaging data. The studies of BFN based on neuroimaging data such as fMRI are significant for the analysis and study of neurological diseases [4,5,6].

The fMRI-based BFN are constructed such that brain regions or voxels are used as nodes and the correlations of BOLD changes between them are used as connection conditions, where most of them are static brain functional networks [7,8]. Static networks can reflect the connectivity pattern of the brain, but ignore the timing change information of BOLD signals; however, with the development of temporal graphs, dynamic brain functional networks (D-BFN) gradually enter the field of view of related researchers and have received more and more attention.

In the research of D-BFN, Wang et al. [9] explored the association between alterations in the dynamic brain networks’ trajectory and cognitive decline in the AD spectrum; the results show that all resting-state networks (RSNs) had an increase in connectivity within networks by enhancing inner cohesive ability, while 7 out of 10 RSNs were characterized by a decrease in connectivity between networks, which indicated a weakened connector among networks from the early stage to dementia. Jie et al. [10] defined a new measure to characterize the spatial variability of DCNs, and then proposed a novel learning framework to integrate both temporal and spatial variabilities of DCNs for automatic brain disease diagnosis. The results of a study on 149 subjects with baseline rs-fMRI data from the Alzheimer’s disease neuroimaging initiative (ADNI) suggest that the method can not only improve the classification performance, but also provide insights into the spatio-temporal interaction patterns of brain activity and their changes in brain disorders. Aldo et al. [11] proposed that spatial maps constituting the nodes in the functional brain network and their associated time-series were estimated using spatial group independent component analysis and dual regression; whole-brain oscillatory activity was analyzed both globally (metastability) and locally (static and dynamic connectivity). Morin et al. [12] used a dynamic network analysis of fMRI data to identify changes in functional brain networks that are associated with context-dependent rule learning; the results support a framework by which a stable ventral attention community and more flexible cognitive control community support sustained attention and the formation of rule representations in successful learners. Moguilner et al. [13] proposed data-driven machine learning pipeline based on dynamic connectivity fluctuation analysis (DCFA) on RS-fMRI data from 300 participant; the results show that non-linear dynamical fluctuations surpass two traditional seed-based functional connectivity approaches and provide a pathophysiological characterization of global brain networks in neurodegenerative conditions (AD and bvFTD) across multicenter data. Wang et al. [14] investigated the effects of driving fatigue on the reorganization of dynamic functional connectivity through our newly developed temporal brain network analysis framework. The method provides new insights into dynamic characteristics of functional connectivity during driving fatigue and demonstrate the potential for using temporal network metrics as reliable biomarkers for driving fatigue detection. Different from conventional studies focusing on static descriptions on functional connectivity (FC) between brain regions in rs-fMRI, recent studies have resorted to D-BFN to characterize the dynamic changes of FC, since dynamic changes of FC may indicate changes in macroscopic neural activity patterns in cognitive and behavioral aspects.

The D-BFNs are usually established by using the sliding window method, which is used to segment the BOLD signals into small pieces—this leads to the size of the window and the length of the split signal directly affecting the effectiveness of the D-BFN. Moreover, due to the sampling point limitation of the BOLD signals, the number of windows used to segment the signals will not be too large, so there is no way to establish the instantaneous BFN of each data segment well. This will lead to the D-BFN established by the sliding window method being discrete. Moreover, because there is a lack of transition between network snapshots, some changing trend information cannot be displayed. The problem that must be faced with the D-BFN based on fMRI data is that the BOLD signal in the data will not have a long duration, which is determined by the characteristics of high spatial resolution and general temporal resolution of fMRI data [15,16]. To establish a D-BFN that can clearly reflect the dynamic changes of brain connectivity, it is necessary not only to consider how to establish a continuous BFN model that is continuous within the time range of data acquisition, but also to extend more time sequence information on the basis of existing time sequence data [17].

Dynamic networks are widely used in social network analysis [18,19], recommendation systems, epidemiology, and others. Representing a complex network as a time-dependent structure allows the network model to exploit not only structural pattern but also temporal patterns. Learning continuous-time dynamics on complex networks is essential for understanding, predicting, and controlling complex systems in science and engineering. Nevertheless, this task is very challenging due to the combinatorial complexity of high-dimensional system structures, the elusive continuous-time nonlinear dynamics, and their structural dynamics dependence. The brain functional network model based on continuous time should consider how to simulate the complete temporal dynamics system by using the existing image data. To meet these challenges, appropriate methods are needed to learn continuous temporal dynamics on complex brain functional networks in a data-driven manner. In recent years, graph neural networks (GNN) have attracted much attention for their excellent performance in a series of network scientific tasks such as link prediction and node classification. Despite the fact that the graph neural network is very popular and the dynamic network models have demonstrated its advantages, little attention is paid to graph neural networks used for dynamic networks. Ordinary differential equation systems (ODEs) [20] area one of the most important mathematical tools used to generate models in physical, biological, chemical, and engineering fields, among others. This prompts researchers to provide effective numerical methods to solve such equations. Higher-order ordinary differential equations are commonly used to solve time-series problems; therefore, one may consider combining ordinary differential equation systems and graph neural networks to learn to compute continuous temporal dynamics on BFNs in a data-driven manner.

D-BFN is usually established by using time window partition to segment BOLD signals, which leads to the size of window and the length of split signals directly affecting the effectiveness of dynamic brain function network [13,21]. Moreover, due to the limitation of sampling points of BOLD signal, the number of divided windows will not be too significant, so there is no way to establish the instantaneous brain function network of each data segment, and it is difficult to predict the change trend of BFN in the next stage after data acquisition.

As there has been no research to consider the dynamic continuity of the network in the establishment of the model of D-BFN, this paper attempts to calculate the continuity information. There are some deep learning methods that can establish the time dynamic response on the network [22], among which the method of Zang et al. [23] is outstanding. The neural dynamics on complex networks (NDCN) model proposed by Zang et al. [23] combines ordinary differential equation systems (ODEs) and graph neural networks (GNNs) to learn continuous-time dynamics on complex networks in a data-driven way. In the view of Skarding et al. [22], there is, as of yet, no continuous DGNN encoder for any general-purpose dynamic network; however, while this approach to modeling dynamics has been discussed by earlier works, to the best of our knowledge, it has not been implemented in practice. For this reason, we discuss the possibility of using the NDCN method in the study of D-BFN.

An extensible dynamic brain function network model is established by using the NDCN. NDCN integrates the GNN layer numerically in continuous time, so as to capture the continuous-time dynamics on the network. This method mainly includes two important functions. One is to calculate the network structure of each instantaneous in the dynamic network within the BOLD signal length by using the existing data, that is, interpolation prediction. The other is to predict the continuous changes of BOLD signal collected by dynamic brain function network in the next time, that is, extensible prediction.

The contributions can be summarized as follows:1The NDCN-Brain gives the meaning of the continuous-time network dynamics to the depth and hidden outputs of GNNs, respectively, and predicts continuous-time dynamics on BFN.2An extended dynamic brain function network model structure is established, which makes up for the length limitation of fMRI data and improves the temporal resolution of the network in a certain range.3The network is applied to the auxiliary diagnosis of cognitive impairment and high diagnostic performance is obtained.

The remainder of the paper is organized as follows, the method used in the model is introduced in Section 2, and the model is tested in Section 3. Finally, the paper is summarized in Section 4.

## 2. Methods

### 2.1. Overview

An extended dynamic brain function network based on neural dynamics on complex networks (NDCN) is proposed as shown in Figure 1.

For the D-BFN model described in this study, the BOLD signal is divided into several slices by window partition, and then the network within each segment is calculated. Then, in-snapshots within the signal length are obtained by using these segment networks through NDCN interpolation prediction, and out-snapshots outside the signal are predicted by extrapolation prediction. These in-snapshots and out-snapshots together constitute a more complex dynamic brain function network model: NDCN-Brain. Using this network dynamics analysis method, we can effectively capture the instantaneous network structure that cannot be obtained by window partition, and predict the network with unknown signal space. Finally, the D-BFN model was tested. Most of the snapshots (including in and out) in the established BFN maintain the conventional attributes of the BFN well in the test. The model was used to classify and predict cognitive impairment data such as Alzheimer’s (AD), early mild cognitive impairment (EMCI), late mild cognitive impairment (LMCI), and normal control (NC) through dynamic network features. It was found that the proposed NDCN-Brain model has a better classification effect than the static network model and the conventional dynamic network model in terms of classification.

As shown in Figure 1 our method is as follows. First, the fMRI data are preprocessed, the preprocessed data are registered into the brain template of power264, and then the BOLD signals of 264 brain regions corresponding to the brain regions are obtained; then, using time window interval sampling, a plurality of signal segments are collected in BOLD signals. The adjacency matrices in each signal segment are calculated, and then the sequential adjacency matrices are brought into the NDCN to obtain the continuous-time dynamics time system. The interpolation prediction is used to analyze every instantaneous snapshots (i.e., in-snapshots) in the dynamic time system, and the extrapolation prediction is used to predict multiple snapshots (i.e., out-snapshots) in the future. These snapshots are combined to obtain an extended dynamic brain functional network structure (in-snapshots, out-snapshots). We can use this model to diagnose cognitive impairment. In the diagnosis process, dynamic network features are extracted first and then diseases are classified based on SVM.

### 2.2. The Brain Functional Network Based on NDCN

In this paper, the neural dynamics on complex networks (NDCN) method proposed by Zang et al. [23] is used to construct a dynamic extensible BFN. This method combines ordinary differential equation systems (ODEs) and graph neural networks (GNNs) to analyze the temporal dynamics of the network in a data-driven manner. Because the brain functional network can also be described as a continuous dynamic network structure in time, the NDCN method can be used to construct the network.

#### 2.2.1. Neural Dynamics on Complex Networks

In the theory of Zang et al. [23], a graph structure with dynamic continuity time can be described as:(1)$$\frac{dX(t)}{dt}=f\left(X,G,W,t\right)$$

In which $X\left(t\right)\in R^{n\times d}$ can be expressed as *n* interconnected nodes in a dynamic continuous system. Each node is characterized by *d* dimension. G=(V,E) represents a network structure that captures how nodes interact. W(t) is a parameter that controls how the system evolves with time. $X\left(0\right)=X_{0}$ is the initial state of the system at time t=0. The equation $f:R^{n\times d}\to R^{n\times d}$ represents the instantaneous change rate of dynamics on the graph. In addition, nodes may have various semantic tags $Y\left(X,\Theta \right)\in {\left(0,1\right)}^{n\times k}$, which encoded by a hot code, and parameter Θ represents this classification function.

The basic framework of NDCN can be summarized as follows:(2)$$\begin{array}{cc} \; & \underset{W(t),\Theta (T)}{\arg\min L}=\int_{0}^{T}R\left(X,G,W,t\right)dt+S\left(Y\left(X\left(T\right),\Theta \right)\right)\\ \; & \;\mathrm{subject}\;\mathrm{to}\;\frac{dX(t)}{dt}=f\left(X,G,W,t\right),X\left(0\right)=X_{0} \end{array}$$
where $\int_{0}^{T}R\left(X,G,W,t\right)dt$ is the “running” loss of the continuous time dynamics on graph from t=0 to t=T. S(Y(X(T),Θ)) is the “terminal” loss at time *T*. By integrating $\frac{dX}{dt}=f\left(X,G,W,t\right)$ over time *t* from initial state $X_{0}$, also known as solving the initial value problem for this differential equation system, can obtain the continuous-time network dynamics $X\left(t\right)=X\left(0\right)+\int_{0}^{T}f\left(X,G,W,\tau \right)$ at arbitrary time moment t>0.

Moreover, to further increase the express ability of our model, NDCN encodes the network signal X(t) from the original space to $X_{h}\left(t\right)$ in a hidden space, and learn the dynamics in such a hidden space. Then, the NDCN model becomes:(3)$$\begin{array}{cc} \; & \underset{W(t),\Theta (T)}{\arg\min}L=\int_{0}^{T}R\left(X,G,W,t\right)dt+S\left(Y\left(X\left(T\right),\Theta \right)\right)\\ \; & \;\mathrm{subject}\;\mathrm{to}\;X_{h}\left(t\right)=f_{e}\left(X\left(t\right),W_{e}\right),X\left(0\right)=X_{0}\\ \; & \;\;\;\;\frac{dX(t)}{dt}=f\left(X_{h},G,W_{h},t\right)\\ \; & \;\;\;\;X\left(t\right)=f_{d}\left(X_{h}\left(t\right),W_{d}\right) \end{array}$$
where the first constraint transforms X(t) into hidden space $X_{h}\left(t\right)$ through encoding function $f_{e}$. The second constraint is the governing dynamics in the hidden space. The third constraint decodes the hidden signal back to the original space with decoding function $f_{d}$. The design of $f_{e}$, *f*, and $f_{d}$ is flexible and can adapt to any deep neural structure, of which GNN is the best. NDCN can realize prediction in two directions, namely interpolation prediction and extrapolation prediction. For one system (0.T], at time t<T, NDCN predicts the prediction result of instantaneous network in continuous time system, that is, interpolation prediction, which is named ’in-snapshots’. At time t>T, NDCN has knowledge of network dynamics outside the system, that is, the prediction results of extrapolated prediction, which is named ’out-snapshots’.

#### 2.2.2. Dynamic Network Modeling Based on NDCN

Before constructing a continuous network, the original discrete-time system needs to be sampled and used for the learning of the continuous-time system. For the fMRI-based brain function network, the BOLD signals are sampled from front to back by the time window partition method. The time series of each brain region is segmented. Specifically, for fMRI data with BOLD signal length *T*, the whole time series (m is the number of nodes) $BOLD_{i}\left(i=1,2,\cdots ,m\right)$ is divided into several time windows with length *t*, and the interval between each window is *s*, that is, the moving step is *s*. Finally, *n* time segments are obtained, and each time period obtained is called a window time. The formula for calculating the total number of windows generated by the time window method is as follows.
(4)$$n=\frac{T-t}{s}+1$$

Figure 2 shows a schematic diagram of the sliding time window, where *T* is the length of the fMRI time series, *t* is the size of the sliding time window, and *s* is the step size of the sliding time window movement.

The Pearson correlation coefficient [24] is used to calculate the correlation between BOLD signals in each time window, and the calculation method is as follows:(5)$$r=\left|\frac{N\sum x_{i}y_{i}-\sum x_{i}\sum y_{i}}{\sqrt{N\sum x_{i}^{2}-{\left(\sum x_{i}\right)}^{2}}\sqrt{N\sum y_{i}^{2}-{\left(\sum y_{i}\right)}^{2}}}\right|$$
where $x_{i},y_{i}$ represents two brain regions in each time slice. Pearson correlation coefficient was calculated in each slice. Then, we obtain the correlation matrix $X\hat{(t})$, that is, an original snapshot of the network. We sample *n* snapshots in a continuous-time system $\left\{X\left({\hat{t}}_{1}\right),\ldots ,X\left(t_{n}\right)|0\leq t_{1}<\ldots <t_{n}\leq T\right\}$ for NDCN training. After the training is completed, for any time i<T, we can use interpolation prediction to obtain in-snapshots X(i); as for the model for j>T, the out-snapshot can be obtained by extrapolation prediction X(j), and finally all the in-snapshot and out-snapshot form an extended dynamic brain function network model with model time dimension > t. This model [X(1),…,X(i),…X(T),…,X(j),…] can be called NDCN-Brain.

### 2.3. Cognitive Impairment CAD Based on NDCN-Brain

Computer-aided diagnosis (CAD) [25,26] generally includes two parts, namely feature extraction and classification. Here we use dynamic aggregation coefficient as the feature of dynamic network and then use support vector machines (SVM) to classify the feature data.

It is known that the brain usually performs multiple functions in the form of partitions, and even general physiological activities will be completed in a multi-regional cooperative way, which is also reflected in the brain network. Because the aggregation coefficient is used to describe the degree of aggregation between vertices in a graph, the global aggregation coefficient can give an evaluation of the aggregation degree of the whole graph; therefore, for a dynamic network structure, each snapshot can give a global aggregation coefficient as a feature, and these features can form an aggregation coefficient sequence, which is used as dynamic network features for classification.

For a general graph structure, that is, a snapshot $G_{snapshot}=\left(V,E\right)$ in this paper, where $V=\left\{v_{1},v_{2},\ldots ,v_{n}\right\}$ represents the set of vertices, $E=\left\{e_{ij}:\left(i,j\right)\in S\subset {\left[1,\cdots ,n\right]}^{2}\right\}$ denotes the set of edges, and $e_{ij}$ denotes the connection of vertices $v_{i}$ and $v_{j}$.

Each vertex is connected with a different number of other vertices, and L(i) is used to represent the set of edges connected with vertex $v_{i}$:(6)$$L\left(i\right)=\left\{v_{j}:e_{ij}\in E\wedge e_{ji}\in E\right\}$$

The number of edges in L(i) is the degree of vertex $v_{i}$, denoted as $k_{i}:k_{j}=\left|L\left(i\right)\right|$.

If $G_{total}^{s}$ is used to denote the global aggregation coefficient of snapshot, $G_{▵}$ is used to denote the number of closed three-point groups in the graph, and $G_{\wedge }$ is used to denote the number of open three-point groups, then
(7)$$G_{total}^{s}=\frac{3\times G_{▵}}{3\times G_{▵}+2\times G_{\wedge }}$$

If it is expressed in $k_{i}$, it can also be written as
(8)$$G_{total}^{s}=\frac{3\times G_{▵}}{\sum_{i=1}^{n}\left(\begin{array}{c} k_{i}\\ 2 \end{array}\right)}$$

For people with cognitive impairment or the extended dynamic brain functional network model [X(1),...,X(i),...X(T),...,X(j),...], the feature vector of aggregation coefficient can be expressed as $[G_{total}^{1},...,G_{total}^{i},...G_{total}^{T},...,G_{total}^{j},\ldots ]$.

Then, support vector machines (SVM) are used to classify the feature vectors of different cognitive impairments, which can realize the auxiliary diagnosis of cognitive impairment diseases.

## 3. Results

### 3.1. Experimental Data and Preprocessing

Alzheimer’s disease data are sourced from the Alzheimer’s Disease Neuroimaging Initiative database http://ADNI.oni.usc.edu/ (accessed on 18 September 2021). The data obtained include Alzheimer’s (AD), early mild cognitive impairment (EMCI), late mild cognitive impairment (LMCI), and normal control (NC).

ADNI provides ethics statements for AD, EMCI, LMCI, and NC. For comparison, 95 samples were reserved in each group. Table 1 shows the detailed data.

For each subject, the resting state fMRI DPARSF toolbox [27] and statistical parameter mapping software package (SPM12) [28] were used to preprocess the data, and the standard preprocessing was applied to the rs-fMRI data set. First, slice correction was carried out. The fMRI time series was recalibrated using spatial transformations to compensate for the effects of head motion. All images are normalized to MNI (Montreal Neurological Institute) space and resampled to 3 mm voxels. Bandpass filtering (0.01–0.2 Hz) was used to preserve valid information.

### 3.2. Network Analysis

In order to verify the effectiveness of the network, the small world attribute was used as the verification method of each snapshot. In previous studies, it has been found that the human brain is a complex network with efficient “small world” attributes [29,30]. If the brain network is in its normal working state, the network mode under most time snapshots of the dynamic network follows the small world attribute. For the in-snapshots within <T time and out-snapshots outside >T time in the model, we verified whether the small world attribute was missing, and compared the differences between different people from cognitive impairment groups.

The shortest path length of the small world network is smaller than that of the regular network, and the clustering coefficient of the small world network is larger than that of the random network. Studies have shown that brain functional networks belong to small world networks, so the evaluation of brain functional networks can be described by the small world attributes of brain networks, and can also reflect the activity ability of parts or whole regions of the brain.

In order to judge the small world characteristics, the characteristic path length *L* and aggregation coefficient *C* are compared with the same measures estimated in random networks. The number of nodes, average degree, and degree distribution in random networks are the same as those in interested networks.

Random networks are expressed as $H_{L}^{R},H_{I}^{R},H_{H}^{R}$. The clustering coefficient is expressed as $C_{L}^{R},C_{I}^{R},C_{H}^{R}$. The shortest path is represented as $L_{L}^{R},L_{I}^{R},L_{H}^{R}$. It is usually expected in a small world network that $\gamma =C/C^{R}>1$ and $\lambda =L/L^{R}>1$. The characteristics of the small world are σ=γ/λ, which are usually greater than 1 (σ>1).

In addition, a plurality of signals corresponding to time positions were intercepted from the original BOLD signals as comparison signals, and $\ell_{1}$ loss and $\ell_{2}$ loss was used as comparison to verify the similarity with the actual signals. For position *i*, a network composed of BOLD signals with a starting time [i+3,i−3] and a length the same as the time window length intercepted during training was used for comparison.

#### 3.2.1. In-Snapshots Analysis

In the process of training, from the beginning of the signal, a number of BOLD signals were obtained with five sampling points as the step size and six sampling points as the signal length; an instantaneous network in every window was established for the training of the model. The instantaneous snapshots in the signal time range were extracted from the acquired model and the usability of these in-snapshots was tested. The test consists of three parts: small world attribute change, signal similarity measure, and the influence of sampled signal window size.

The first were the $\ell_{1}$ loss and $\ell_{2}$ loss. In fMRI data, a volume is a time point. According to the length of the data signal, we can obtain 140 in-snapshots in the model. For primitive in-snapshots *i* (predicted value), a network composed of BOLD signals with a [i+3,i−3] time window length intercepted during training (true value) is used for comparison. The comparison results are shown in Table 2; means and errors of all the same subjects are shown in the table. The table mainly compares the error between the predicted value of NDCN model and the network partitioned by traditional windows. In addition, some other deep learning methods are compared, we use GNN [31] as a graph structure extractor and use LSTM/GRU/RNN [32] to learn the temporal relationships between ordered structured sequences.

It can be seen from the table that the proposed NDCN method, whether $\ell_{1}$ loss and $\ell_{2}$ loss, NDCN obtains the smallest error value.

In addition, it is necessary to compare the influence of sliding window size on the obtained results. It can be seen from Figure 3 and Figure 4, that when the size of sliding window is 6, the loss is the smallest; all experiments were carried out on the basis of this window size.

After that, it was necessary to verify whether every time segment in the dynamic network obtained by the NDCN model still had the attribute of a small world, which is an important basis for the usability of the model. In Table 3, no matter what cognitive impairment, their small world attributes were preserved.

In the experiment, it was found that most snapshots retained the small world attribute, but the proportion was different among different people with diseases. According to the experimental results, it is found that the more serious the cognitive impairment is, the more snapshots lose the small world attribute in their dynamic network. Among them, only 70.7% of the fragments in the dynamic network of AD patients retain the small world attribute. Healthy people have the largest number, maintaining 92.1%.

#### 3.2.2. Out-Snapshots Analysis

The instantaneous snapshots outside the signal time range were extracted from the obtained model and the usability of these out-snapshots was tested; 70 out-snapshots were calculated, which also includes two parts: small world attribute change and signal similarity measurement. The same comparison method as in-snapshots was adopted. The $\ell_{1}$ loss and $\ell_{2}$ loss are shown in Table 4.

It can be seen that, compared to other deep learning methods, out-snapshots using the NDCN model also feature small errors.

After that, it is still necessary to verify whether every time segment in the dynamic network obtained by NDCN model still has the attribute of small world, which is an important basis for the usability of the model. In Table 5, no matter what people with cognitive impairment, their small world attributes have been preserved.

In the experiment, it was found that most out-snapshots retained the small world attribute, but the proportion was different among people with different diseases. According to the experimental results, it is also found that the more serious the cognitive impairment is, the more snapshots lose the small world attribute in their dynamic network. Among them, only 65.7% of the fragments in the dynamic network of AD patients retain the small-world attribute. Healthy people have the largest number, maintaining 88.6%.

#### 3.2.3. Comparison with Continuous Dynamic Network Models

In the existing research, there have been some methods that can be used to calculate the continuous time dynamics model, but these calculation methods have unique pertinence and are not fully applicable to the brain function network; therefore, we compare some methods here to prove the superiority of the NDCN-Brain method, such as RNN-based, TTP-based, and time-embedding-based models by NC subjects.

It can be seen from Table 6 that NDCN model is more suitable for the modeling of continuous dynamic brain function network.

**Table 6 diagnostics-12-01298-t006:** Comparison with continuous dynamic network models.

Model Type	Model Name	$\ell_{1}$ Loss	$\ell_{2}$ Loss
RNN-based	Streaming GNN [33]	34.6 ± 4.2	22.8 ± 3.9
	JODIE [34]	35.3 ± 2.8	23.6 ± 3.1
TTP-based	Know-Evolve [35]	38.6 ± 5.4	28.1 ± 2.8
	DyREP [36]	33.7 ± 4.1	42.6 ± 2.2
	LDG [37]	41.6 ± 3.4	27.5 ± 5.3
	GHN [38]	33.4 ± 4.2	25.9 ± 4.5
Time-embedding-based	TGAT [39]	33.7 ± 2.7	26.5 ± 3.8
	TGN [40]	29.6 ± 4.2	16.7 ± 3.6
NDCN	NDCN-brain	25.7 ± 3.7	12.8 ± 3.7

#### 3.2.4. Auxiliary Diagnosis for Cognitive Impairment

The purpose of ADNI is to develop a multisite, longitudinal, prospective, and naturalistic study of normal cognitive aging, mild cognitive impairment (MCI), and early Alzheimer’s disease as a public domain research resource to facilitate the scientific evaluation of neuroimaging and other biomarkers for the onset and progression of MCI and Alzheimer’s disease (AD) [41]. Early mild cognitive impairment (EMCI), late mild cognitive impairment (LMCI), and AD are different stages in the development of AD (NC→EMCI→LMCI→AD). Identifying different stages is helpful to understand the characteristic changes of brain network in corresponding stages, and provide reference for disease research and prevention; therefore, the proposed method is used to distinguish the stages of Alzheimer’s development.

Auxiliary diagnosis mainly uses SVM to classify the features. The network structure includes in-snapshots and out-snapshots. Here, the aggregation coefficient of network $[G_{total}^{1},\ldots ,G_{total}^{i},\ldots G_{total}^{T},\ldots ,G_{total}^{j},\ldots ]$ was used as the feature. In the experimental comparison, the traditional window partition method was used as the baseline. Three classifiers, KNN, naive Bayesian (NB), and SVM were compared [42,43]. The experimental results are shown in Table 7.

Classification mainly focuses on the distinction between cognitive impairment and normal population. From the experimental results, it can be seen that the classification effect for Alzheimer’s is the best, which is 82.5%, while late cognitive impairment and early cognitive impairment are second (77.5% and 72.5%). This is because the more serious the cognitive impairment, the higher the discrimination of its dynamic network. Finally, we can see that the dynamic network of NDCN model has certain advantages compared with direct window partition. It can not only use the information of BOLD signal, but also predict the network information in the future, and integrate its characteristics for classification.

In addition, the influence of in-snapshots and out-snapshots on classification is verified. Here, the classifier only uses SVM. In Table 8, the baseline is the method of window partition, and the results of in-snapshots and out-snapshots are compared, respectively. The experimental results show that the NDCN method has the best diagnostic effect.

It can be inferred that when the degree of cognitive impairment deepens, the accumulation of brain structure decreases, and the network in the normal brain will show higher accumulation. The accumulation of cognitive impairment patients will become worse, and the realization of function will be limited. This is why clustering coefficient can be used as a feature for classification. In addition, because the dynamic network construction method used in this paper is no longer directly practical time window, but increases the inference of network timing changes, and the information is retained more completely, so it has better classification performance.

## 4. Conclusions

In order to solve the problems of instantaneous network extraction and network development prediction outside the signal, an extensible dynamic brain function network model is proposed. The model utilizes the ability of extracting and predicting the instantaneous state of the dynamic network of NDCN and constructs a dynamic network model structure that can provide more than the original signal range. Experimental results show that every snapshot in the network obtained by the proposed method has a usable network structure and it also has a good classification result in the diagnosis of cognitive impairment diseases.

The method proposed in this paper calculates the time dynamic response of brain function network. There is still some room for improvement. The model calculates the internal characteristics of the network, but this feature is only used to calculate the time dynamic response, while the feature used in disease classification is the agglomeration coefficient after extracting time segments, which cannot fully reflect the changes of network connection under each time slice. In addition, by changing the loss function, the characteristic information of the network can be further extracted. If the connection change information of network nodes can be extracted, the model can also be used in the research of network integration and separation, which will be verified by further experiments. The model only analyzes the brain network of Alzheimer’s disease and its application in other diseases needs to be developed. 

## Figures and Tables

**Figure 1 diagnostics-12-01298-f001:**
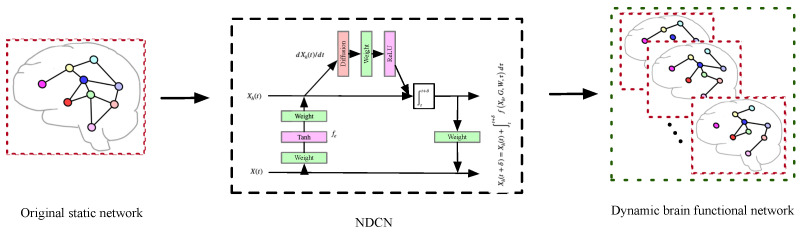
Overview of dynamic brain function network based on NDCN.

**Figure 2 diagnostics-12-01298-f002:**
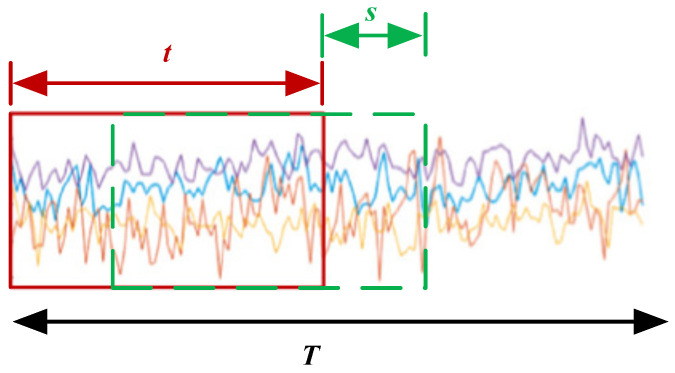
Sliding time window schematic diagram.

**Figure 3 diagnostics-12-01298-f003:**
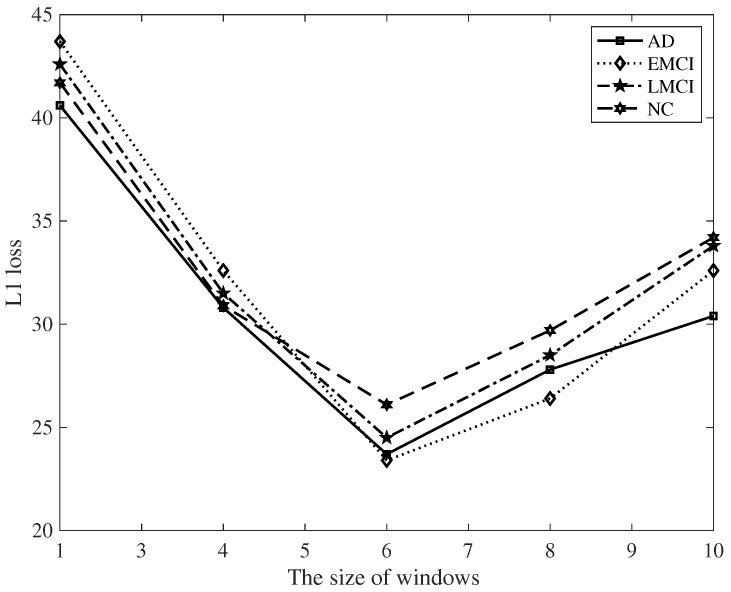
Influence of sliding window on $\ell_{1}$ loss.

**Figure 4 diagnostics-12-01298-f004:**
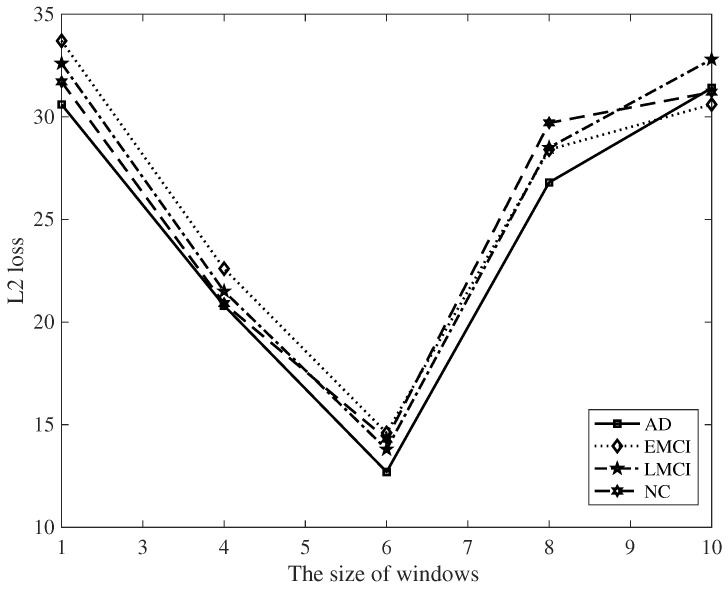
Influence of sliding window on $\ell_{2}$ loss.

**Table 1 diagnostics-12-01298-t001:** Details of ADNI data set.

Dataset	Subjets (Male/Female)	Ages	Slices	TE	TR	No. TRs
AD	47/48	73.6 ± 15	3.31 mm	30 ms	3.0 s	140
EMCI	47/48	72.7 ± 12	3.31 mm	30 ms	3.0 s	140
LMCI	47/48	74.8 ± 14	3.31 mm	30 ms	3.0 s	140
NC	47/48	75.1 ± 13	3.31 mm	30 ms	3.0 s	140

**Table 2 diagnostics-12-01298-t002:** The $\ell_{1}$ loss and $\ell_{2}$ loss comparison results of in-snapshots.

Dataset	LSTM-GNN		GRU-GNN		RNN-GNN		NDCN-Brain	
	$\ell_{1}$ Loss	$\ell_{2}$ Loss	$\ell_{1}$ Loss	$\ell_{2}$ Loss	$\ell_{1}$ Loss	$\ell_{2}$ Loss	$\ell_{1}$ Loss	$\ell_{2}$ Loss
AD	51.8 ± 3.1	33.1 ± 3.6	44.8 ± 3.0	35.3 ± 2.7	43.1 ± 3.4	32.6 ± 4.2	23.7 ± 3.6	12.7 ± 4.4
LMCI	50.7 ± 2.8	34.4 ± 2.4	47.3 ± 2.8	32.6 ± 3.6	42.9 ± 3.6	31.4 ± 4.5	23.4 ± 2.8	14.6 ± 2.7
EMCI	51.5 ± 3.4	36.7 ± 2.8	43.5 ± 3.3	31.8 ± 3.3	46.1 ± 3.3	29.7 ± 4.6	24.5 ± 4.3	13.8 ± 3.3
NC	49.8 ± 2.4	37.9 ± 2.9	42.7 ± 3.6	37.1 ± 2.6	41.7 ± 3.2	33.2 ± 4.3	26.1 ± 4.7	14.3 ± 4.2

**Table 3 diagnostics-12-01298-t003:** Average small world results.

Dataset	$\bar{C}$	$\bar{L}$	${\bar{C}}^{R}$	${\bar{L}}^{R}$	$\bar{\gamma }$	$\bar{\lambda }$	$\bar{\sigma }$	σ>1
AD	0.6834	1.8903	0.3947	1.7542	1.7314	1.0775	1.6067	70.7%(99/140)
LMCI	0.6543	1.7654	0.3876	1.6455	1.6880	1.0728	1.5734	83.5%(117/140)
EMCI	0.5874	1.8763	0.4322	1.7355	1.3590	1.0811	1.2571	87.8%(123/140)
NC	0.6446	1.9976	0.3872	1.7365	1.6647	1.1503	1.4471	92.1%(129/140)

**Table 4 diagnostics-12-01298-t004:** The $\ell_{1}$ loss and $\ell_{2}$ loss comparison results of out-snapshots.

Dataset	LSTM-GNN		GRU-GNN		RNN-GNN		NDCN-Brain	
	$\ell_{1}$ Loss	$\ell_{2}$ Loss	$\ell_{1}$ Loss	$\ell_{2}$ Loss	$\ell_{1}$ Loss	$\ell_{2}$ Loss	$\ell_{1}$ Loss	$\ell_{2}$ Loss
AD	52.6 ± 3.3	31.7 ± 3.4	43.2 ± 2.8	34.3 ± 2.2	44.2 ± 3.5	32.7 ± 4.3	22.6 ± 3.4	14.8 ± 4.2
LMCI	51.2 ± 2.4	32.1 ± 3.3	45.2 ± 3.1	34.2 ± 3.4	43.6 ± 3.4	31.8 ± 4.1	22.3 ± 2.5	13.9 ± 4.7
EMCI	49.5 ± 3.2	33.6 ± 3.1	45.3 ± 2.8	33.6 ± 3.4	42.2 ± 3.7	28.7 ± 4.3	27.4 ± 3.3	14.6 ± 4.3
NC	52.0 ± 2.3	32.4 ± 3.9	43.6 ± 2.6	32.2 ± 3.6	43.1 ± 3.5	29.2 ± 4.1	25.7 ± 3.7	12.8 ± 3.7

**Table 5 diagnostics-12-01298-t005:** Average small world results.

Dataset	$\bar{C}$	$\bar{L}$	${\bar{C}}^{R}$	${\bar{L}}^{R}$	$\bar{\gamma }$	$\bar{\lambda }$	$\bar{\sigma }$	σ>1
AD	0.6907	1.7532	0.4533	1.6732	1.5237	1.0478	1.4541	65.7%(46/70)
LMCI	0.6343	1.7344	0.3673	1.5433	1.7269	1.1238	1.5366	75.7%(53/70)
EMCI	0.5874	1.5673	0.3576	1.5312	1.6426	1.0235	1.6047	84.3%(59/70)
NC	0.6456	1.834	0.2398	1.6322	2.6922	1.1236	2.3960	88.6%(62/70)

**Table 7 diagnostics-12-01298-t007:** Auxiliary diagnosis for cognitive impairment.

Dataset	Classifier	Method	Accuracy %	Sensitivity %	Specificity %	AUC
AD vs. NC	KNN	Baseline	65.0	70.0	60.0	0.65
		NDCN-brain	79.5	82.0	79.0	0.81
	NB	Baseline	67.5	75.0	60.0	0.67
		NDCN-brain	81.4	81.0	82.5	0.84
	SVM	Baseline	70.0	75.0	65.0	0.69
		NDCN-brain	82.5	85.0	80.0	0.87
LMCI vs. NC	KNN	Baseline	62.5	75.0	50.0	0.59
		NDCN-brain	74.5	78.0	73.0	0.78
	NB	Baseline	60.0	65.0	55.0	0.63
		NDCN-brain	76.5	77.5	73.0	0.75
	SVM	Baseline	62.5	70.0	55.0	0.63
		NDCN-brain	77.5	80.0	75.0	0.78
EMCI vs. NC	KNN	Baseline	57.5	60.0	55.0	0.56
		NDCN-brain	71.5	72.5	65.0	0.70
	NB	Baseline	60.0	60.0	60.0	0.57
		NDCN-brain	70.5	75.0	70.0	0.74
	SVM	Baseline	62.5	75.0	50.0	0.59
		NDCN-brain	72.5	75.0	70.0	0.74

**Table 8 diagnostics-12-01298-t008:** The influence of in-snapshots and out-snapshots.

Dataset	Method	Accuracy %	Sensitivity %	Specificity %	AUC
AD vs. NC	Baseline	70.0	75.0	65.0	0.69
	In-snapshots	77.5	78.0	80.5	0.82
	Out-snapshots	74.5	75.0	78.0	0.78
	NDCN-brain (In + Out)	82.5	85.0	80.0	0.87
LMCI vs. NC	Baseline	62.5	70.0	55.0	0.63
	In-snapshots	72.0	73.0	65.0	0.75
	Out-snapshots	68.5	72.0	68.0	0.70
	NDCN-brain (In + Out)	77.5	80.0	75.0	0.78
EMCI vs. NC	Baseline	62.5	75.0	50.0	0.59
	In-snapshots	68.5	75.0	70.0	0.68
	Out-snapshots	65.5	70.0	65.0	0.64
	NDCN-brain (In + Out)	72.5	75.0	70.0	0.74

## Data Availability

The data are from Alzheimer’s Disease Neuroimaging Initiative database http://ADNI.oni.usc.edu/ (accessed on 18 September 2021).

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
