# Peer review of "NDCN-Brain: An Extensible Dynamic Functional Brain Network Model"

_diagnostics, 2022, doi:10.3390/diagnostics12051298_

Round 1
Reviewer 1 Report
The manuscript was modified very well. The authors have attempted to address all reviewers' comments in the revised paper. The manuscript seems acceptable to me for publication in the journal with the corrections made.
Author Response
Thank you very much for your affirmation.Reviewer 2 Report
The authors propose a dynamic brain function network model for fMRI signal. This is a meaningful study. It can be accepted in present form.
Author Response
Thank you very much for your affirmation.
Reviewer 3 Report
In their study, "NDCN-Brain: An Extensible Dynamic Functional Brain Network Model", the authors provide a new method for measuring continuous time dynamics of fMRI brain network connectivity based from complex network theory. To compute continuous temporal dynamics on functional brain networks, the authors develop a model that integrates ordinary differential equation systems with graph neural networks.
The study is interesting because it attempts to investigate a novel approach for measuring fMRI time series that might aid in the early detection of dementia. There is also a growing body of research on the use of functional MRI signal changes to help in the early detection of cognitive impairment in dementia patients. The field is hampered by a scarcity of data to back up the importance of tangible outcomes.
While the research is significant, I have a few major and minor issues that must be addressed. Namely:
1) There are multiple typos in the introduction, and certain sections are difficult to understand. I recommend utilizing abbreviations throughout the document and adhering to them.
2) In the introduction, the authors may add a more detailed (and controversial) overview of current research using fMRI network dynamics analysis for the early diagnosis of neurodegenerative disorders.
3) I suggest better highlighting; what makes the presented model superior? What diagnostic feature would it be able to classify more accurately?
4) I believe the paragraph starting on line 108 belongs in the methods section rather than the introduction.
5) The classification's description The Alzheimer's Disease Neuroimaging Initiative (ADNI) is missing for Alzheimer's disease, Early Mild Cognitive Impairment (EMCI), Late Mild Cognitive Impairment (LMCI), and Normal Control (NC) conditions. This is a crucial component. Because the classification is mostly dependent on identifying cognitive impairment, it's critical to know how cognitive impairment is defined and how this classification approach may help with diagnosis.
6) It would be beneficial to examine further any trend in the correlations between specific cognitive features and fMRI data.
7) Figures 1, 2, 3, and 4 are low quality. "It can be seen from the Figures 3 and 4, that when the sliding window is 6, the error is the smallest" says the explanation for fig. 3 and 4. This is not evident from the figure. It looks that window 4 is the smallest.
8) Line 342-344: "It can be seen from the table 6 that NDCN model is more suitable for the modeling of continuous dynamic brain function network. The choice continuous methods depends on the data and the intended problem." Which data? and what was the intended problem ? Please explain why NDCN is more appropriate and how it can help with early symptom identification and classification.
9) The conclusion could be expanded to include the proposed method's limitations, limitations of the results' generalizability, future directions for improving the method, and diagnostic applications.
10) There are multiple typos in all sections. Please double-check the wording for any mistakes.
Author Response
Point 1: There are multiple typos in the introduction, and certain sections are difficult to understand. I recommend utilizing abbreviations throughout the document and adhering to them.
Response 1: Thank you very much for your comments. We have revised the full text, checked and corrected grammar errors and typos. In addition, according to your suggestions, some nouns use abbreviations in the full text.
Point 2: In the introduction, the authors may add a more detailed (and controversial) overview of current research using fMRI network dynamics analysis for the early diagnosis of neurodegenerative disorders.
Response 2: Thank you very much for your comments. We have added the overview in the introduction. The added part in this revision in section ` Introduction ' is shown as follows:
In the research of D-BFN, Wang et. Al[9] explored the association between alterations in the dynamic brain networks' trajectory and cognitive decline in the AD spectrum, the results shown that all resting-state networks (RSNs) represented an increase in connectivity within networks by enhancing inner cohesive ability, while 7 out of 10 RSNs were characterized by a decrease in connectivity between networks, which indicated a weakened connector among networks from the early stage to dementia. Jie et al[10] define a new measure to characterize the spatial variability of DCNs, and then propose a novel learning framework to integrate both temporal and spatial variabilities of DCNs for automatic brain disease diagnosis. Results on 149 subjects with baseline rs-fMRI data from the Alzheimer's Disease Neuroimaging Initiative (ADNI) suggest that the method can not only improve the classification performance, but also provide insights into the spatio-temporal interaction patterns of brain activity and their changes in brain disorders. Aldo et al[11] proposed that spatial maps constituting the nodes in the functional brain network and their associated time-series were estimated using spatial group independent component analysis and dual regression, and whole-brain oscillatory activity was analyzed both globally (metastability) and locally (static and dynamic connectivity). Morin et al[12] used a dynamic network analysis of fMRI data to identify changes in functional brain networks that are associated with context-dependent rule learning, the results support a framework by which a stable ventral attention community and more flexible cognitive control community support sustained attention and the formation of rule representations in successful learners. Moguilner et al[13] proposed data-driven machine learning pipeline based on dynamic connectivity fluctuation analysis (DCFA) on RS-fMRI data from 300 participant, the results show that non-linear dynamical fluctuations surpass two traditional seed-based functional connectivity approaches and provide a pathophysiological characterization of global brain networks in neurodegenerative conditions (AD and bvFTD) across multicenter data. Wang et al[14] investigated the effects of driving fatigue on the reorganization of dynamic functional connectivity through our newly developed temporal brain network analysis framework. The method provides new insights into dynamic characteristics of functional connectivity during driving fatigue and demonstrate the potential for using temporal network metrics as reliable biomarkers for driving fatigue detection. Different from conventional studies focusing on static descriptions on functional connectivity (FC) between brain regions in rs-fMRI, recent studies have resorted to D-BFN to characterize the dynamic changes of FC, since dynamic changes of FC may indicate changes in macroscopic neural activity patterns in cognitive and behavioral aspects.
Point 3: I suggest better highlighting; what makes the presented model superior? What diagnostic feature would it be able to classify more accurately?
Response 3: Thank you very much for your comments. We have added corresponding explanations in the revised version, such as in section 3.2.4:
It can be inferred that when the degree of cognitive impairment deepens, the accumulation of brain structure decreases, and the network in the normal brain will show higher accumulation. The accumulation of cognitive impairment patients will become worse, and the realization of function will be limited. This is why clustering coefficient can be used as a feature for classification. In addition, because the dynamic network construction method used in this paper is no longer directly practical time window, but increases the inference of network timing changes, and the information is retained more completely, so it has better classification performance.
Point 4: I believe the paragraph starting on line 108 belongs in the methods section rather than the introduction.
Response 4: Thank you very much for your comments. This part has been removed from the introduction and modified into the method.
Point 5: The classification's description The Alzheimer's Disease Neuroimaging Initiative (ADNI) is missing for Alzheimer's disease, Early Mild Cognitive Impairment (EMCI), Late Mild Cognitive Impairment (LMCI), and Normal Control (NC) conditions. This is a crucial component. Because the classification is mostly dependent on identifying cognitive impairment, it's critical to know how cognitive impairment is defined and how this classification approach may help with diagnosis.
Response 5: Thank you very much for your comments. We added the corresponding description in the modified version about ADNI, The added part is as follows:
The purpose of ADNI is to develop a multisite, longitudinal, prospective, naturalistic study of normal cognitive aging, mild cognitive impairment (MCI), and early Alzheimer's disease as a public domain research resource to facilitate the scientific evaluation of neuroimaging and other biomarkers for the onset and progression of MCI and Alzheimer's disease (AD) [41]. Early Mild Cognitive Impairment (EMCI), Late Mild Cognitive Impairment (LMCI), AD are different stages in the development of AD (NC->EMCI->LMCI->AD). Identifying different stages is helpful to understand the characteristic changes of brain network in corresponding stages, and provide reference for disease research and prevention..Therefore, the proposed method is used to distinguish the stages of Alzheimer's development.
Point 6: It would be beneficial to examine further any trend in the correlations between specific cognitive features and fMRI data.
Response 6: Thank you very much for your comments. You have provided a great research direction. We will continue to study the development and change trend of cognitive diseases on the basis of this article.
Point 7: Figures 1, 2, 3, and 4 are low quality. "It can be seen from the Figures 3 and 4, that when the sliding window is 6, the error is the smallest" says the explanation for fig. 3 and 4. This is not evident from the figure. It looks that window 4 is the smallest.
Response 7: Thank you very much for your comments. We have improved the resolution of all images. In addition, I'm sorry we used the wrong description here, In the sentence ‘the error is the smallest’, ‘error’ is ‘loss’, what we want to describe is the lowest point of the curve. Amendments have been made in the new version. Such as ‘that when the size of sliding window is 6, the loss is the smallest’
Point 8: Line 342-344: "It can be seen from the table 6 that NDCN model is more suitable for the modeling of continuous dynamic brain function network. The choice continuous methods depend on the data and the intended problem." Which data? and what was the intended problem? Please explain why NDCN is more appropriate and how it can help with early symptom identification and classification.
Response 8: Thank you very much for your comments. What we originally wanted to express here is that different dynamic corresponding methods should be adopted for different types of data, whether social, traffic or other network data. Since the content has little to do with this article, we have deleted this sentence ‘The choice continuous methods depend on the data and the intended problem.’ in the new version.
Point 9: The conclusion could be expanded to include the proposed method's limitations, limitations of the results' generalizability, future directions for improving the method, and diagnostic applications.
Response 9: Thank you very much for your comments. We added limitations, limitations of the results' generalizability, future directions part to the conclusion. The modified conclusion is as follows:
In order to solve the problems of instantaneous network extraction and network development prediction outside the signal. An extensible dynamic brain function network model is proposed. The model utilizes the ability of extracting and predicting the instantaneous state of the dynamic network of NDCN, and constructs a dynamic network model structure which can provide more than the original signal range. Experimental results show that every snapshots in the network obtained by the proposed method has a usable network structure, and it also has a good classification result in the diagnosis of cognitive impairment diseases.
The method proposed in this paper calculates the time dynamic response of brain function network, and there is still some room for improvement. The model calculates the internal characteristics of the network, but this feature is only used to calculate the time dynamic response, while the feature used in disease classification is the agglomeration coefficient after extracting time segments, which cannot fully reflect the changes of network connection under each time slice. In addition, by changing the loss function, the characteristic information of the network can be further extracted. If the connection change information of network nodes can be extracted, the model can also be used in the research of network integration and separation, which will be verified by further experiments. And the model only analyzes the brain network of Alzheimer's disease, and its application in other diseases needs to be developed.
Point 10: There are multiple typos in all sections. Please double-check the wording for any mistakes.
Response 10: Thank you very much for your comments. We have checked and corrected grammar errors and typos.
Round 2
Reviewer 3 Report
This was a thorough revision and the authors have been very responsive to my feedback.